# Bacterial and Fungal Diversity Inside the Medieval Building Constructed with Sandstone Plates and Lime Mortar as an Example of the Microbial Colonization of a Nutrient-Limited Extreme Environment (Wawel Royal Castle, Krakow, Poland)

**DOI:** 10.3390/microorganisms7100416

**Published:** 2019-10-03

**Authors:** Magdalena Dyda, Adam Pyzik, Ewa Wilkojc, Beata Kwiatkowska-Kopka, Aleksandra Sklodowska

**Affiliations:** 1Research and Development for Life Sciences Ltd., 02-096 Warsaw, Poland; a.pyzik@rdls.pl; 2University of Warsaw, Faculty of Biology, Laboratory of Environmental Pollution Analysis, 02-096 Warsaw, Poland; asklodowska@biol.uw.edu.pl; 3University of Warsaw, Faculty of Biology, Department of Bacterial Genetics, 02-096 Warsaw, Poland; 4Wawel Royal Castle, 31-001 Krakow, Poland; ewa.wilkojc@wawel.org.pl (E.W.); beata.kopka@wawel.org.pl (B.K.-K.)

**Keywords:** microbial diversity, high-throughput sequencing, sandstone, lime mortar, limited nutrient conditions, extreme environment, biodeterioration

## Abstract

Biodeterioration is a serious threat to cultural heritage objects and buildings. The deterioration of a given material often incurs irreparable losses in terms of uniqueness and historical value. Hence preventive actions should be taken. One important challenge is to identify microbes involved in the biodeterioration process. In this study, we analyzed the microbial diversity of an ancient architectonical structure of the Rotunda of Sts. Felix and Adauctus, which is a part of the Wawel Royal Castle located in Krakow, Poland. The Rotunda is unavailable to tourists and could be treated as an extreme habitat due to the low content of nutrients coming either from sandstone plates bound with lime mortar or air movement. Microbial diversity was analyzed with the use of the high-throughput sequencing of marker genes corresponding to fragments of 16S rDNA (for Bacteria) and ITS2 (internal transcribed spacer 2) (for Fungi). The results showed that the microbial community adhered to wall surfaces is, to a large extent, endemic. Furthermore, alongside many microorganisms that could be destructive to masonry and mortar (e.g., *Pseudomonas*, *Aspergillus*), there were also bacteria, such as species of genera *Bacillus*, *Paenisporosarcina*, and *Amycolatopsis*, that can positively affect wall surface properties by reducing the damage caused by the presence of other microorganisms. We also showed that airborne microorganisms probably have little impact on the biodeterioration process as their abundance in the microbial community adhered to the ancient walls was very low.

## 1. Introduction

Cultural heritage buildings and objects are heterogeneous habitats, that can be colonized by various microorganisms. Microbial growth and metabolic activities pose a threat to historical structures due to their material degradation potential. The process is called biodeterioration [1]. The following three main types of biodeterioration processes are known: (1) chemical, caused by organic and inorganic acids and chelating compounds; (2) mechanical, caused by the growth of microorganisms; (3) fouling or soiling as the result of biofilm formation [2,3]. Taxonomic groups shown to be involved in the biodeterioration process include Bacteria, Archaea, Cyanobacteria, Fungi, and Lichens [2,3]. It seems that bacteria and fungi contribute the most to the destruction of valuable historical objects. Bacteria and fungi usually have high growth rates and may tolerate extreme environmental conditions, such as low nutrient concentrations, wide ranges of pH and temperature as well elevated concentrations of compounds that are toxic to other organisms. Moreover, bacteria and fungi are able to relatively easily adjust to changing environmental conditions, especially when they are a part of biofilm. It is assumed that natural stone surfaces are first colonized by chemoorganotrophic or heterotrophic bacteria, which produce organic matter, followed by fungi [4].

Due to the negative impact of microorganisms on cultural heritage there is a need for the identification and exploration of specific microorganisms involved in biodeterioration processes, which in turn may lay the groundwork for the implementation of preventive and corrective measures. In the past, microbial diversity was investigated with culture-dependent methods. However nowadays culture-independent molecular techniques are recommended for routine use. Those methods are based on the direct isolation of total environmental DNA and the subsequent amplification of DNA fragments and amplicon sequencing [5,6]. Commonly used marker genes include 16S rDNA gene for bacteria and ITS2 (internal transcribed spacer 2) for fungi. Whereas amplicon sequencing is relatively easy and cost-effective, it provides useful information about environmental microorganisms forming a complex consortium.

Here, we employed this methodology to investigate microorganisms inhabiting an ancient architectonical structure of the Rotunda of Sts. Felix and Adauctus. The building has been assumed to have an endemic microbial community, as it is closed to tourist traffic and the air ventilation system is limited to the entrance (1.8 m × 1.3 m). The Rotunda is a part of the Wawel Royal Castle located in Krakow, Poland (https://wawel.krakow.pl/en; https://medievalheritage.eu/en/main-page/heritage/poland/krakow-royal-castle/) - one of the most important sites of Polish history and culture. It used to serve as a coronation place and a royal necropolis of Polish kings. 

## 2. Materials and Methods

### 2.1. Site Description

The Rotunda of Sts. Felix and Adauctus is the oldest building of the Wawel Royal Castle complex and is estimated to have been built in the 10th/11th century, when the king of Poland had established his residence at the Wawel Hill for the first time. The Rotunda was directly built on limestone bedrock. An irregular sandstone plate bound with lime mortar was used for construction (Figure 1). The extant fragment of the building wall reaches up to 37 m in height. Above the preserved walls, the Rotunda of Sts. Felix and Adauctus was reconstructed, maintaining its cylindrical character with four apses (https://wawel.krakow.pl/en/exhibition-constant/the-lost-wawel).

### 2.2. Sample Collection

Samples were collected from the Rotunda of Sts. Felix and Adauctus in four months of the year: (1) April, (2) June, (3) August, and (4) October. In each of the sampling periods, the following three types of materials were obtained: (1) sand-like material and dust collected from ancient walls with the use of sterile brushes and (2) air (50 L) collected with the use of a MAS-100 Eco^®^ air sampler (Merck KGaA, Darmstadt, Germany) onto agar plates containing different media and (3) microorganisms collected using sterile swabs and examined using the plate method. Swabs were taken from different sampling points on irregular sandstone plate surfaces from areas of 50 cm^2^. The sampling swabs were immersed in sterile saline (2 mL) in the laboratory tubes, shaken, and spread (0.1 mL) onto agar plates.

Sand-like/dust samples were used for a direct extraction of metagenomic DNA (a culture-independent method), while the air sampled onto agar plates was used for the cultivation of airborne microorganisms, and then the total biomass of colonies was subjected to the extraction of DNA (a culture-dependent method). Microbes were collected using swabs and were further cultivated on agar plates that were used for the siderophore assay only.

### 2.3. Cultivation of Microorganisms

In the cultivation process, six types of standard agar media were used. Three media were dedicated to bacteria, i.e., (1) Nutrient agar, (2) Blickfeldt medium, and (3) Yeast extract agar and the other three were dedicated to fungi, i.e., (4) Czapek Dox agar, (5) Malt extract agar, and (6) Wort agar. Agar media were manufactured by BTL (Poland; www.btl.com.pl). Bacteria and fungi were incubated for 48 h at 37 °C and 7 days at 26 °C, respectively. After cultivation, all colonies grown on bacterial agar plates and those on fungal agar plates from a particular period were collected as two separate samples. The collection of the samples was carried out by washing the cells with sterile saline (3 mL), then transferring into laboratory tubes and centrifuging (8000× *g*, 4 °C, 10 min).

### 2.4. Siderophore Detection Assay

Bacterial cells collected from three different cultivation media and – separately - from a fragment of the sandstone of the Rotunda’s wall were transferred into standard lysogeny broth (LB) medium in order to multiply the cells. Then the cultures were centrifuged (10,000× *g*, 4 °C, 10 min), washed twice in saline, transferred into glucose asparagines medium (GASN) [7], and incubated for 72 h at room temperature. The cultures were then centrifuged (10,000× *g*, 4 °C, 10 min) and the supernatants were used as siderophore preparations after pH adjusting at 6.8 – 7.1. The siderophores in the supernatants were quantified by measuring the absorbance at 630 nm following a 1 h incubation in the Chrome Azurol S (CAS) assay solution [8]. The siderophore concentrations were estimated from a standard curve prepared using deferoxamine mesylate (DFOB) and expressed as mM DFOB [9].

### 2.5. Extraction of Total DNA, Marker Gene Amplifications and High-Throughput Sequencing of 16S rDNA and ITS2 Amplicons

In both cases of DNA extraction from the sand-like/dust sample and the microorganisms cultivated from the air, the isolation was carried out for approximately 0.2 g of a given material with the use of FastDNA™ SPIN Kit for Feces (MP Biomedicals, Solon, OH, USA) and FastPrep^®^ bead beater (MP Biomedicals), according to the manufacturer’s protocols. The metagenomic DNA isolated from both the sand-like/dust samples and the laboratory cultures were used as a template for the amplification of amplicons covering the V3 - V4 hypervariable regions of 16S rDNA gene (for bacteria) and ITS2 fragment (for fungi). The PCR reaction was carried out in triplicate for each of the samples. The amplification process consisted of: (I) initial denaturation at 95 °C for 5 min; (II) 25 cycles of denaturation at 95 °C for 30 s, primer attachment at 60 °C for 30 s, DNA strand synthesis at 72 °C for 30 s; (III) final DNA strand extension at 72 °C for 5 min. The PCR reaction was carried out in a Mastercycler Nexus GX2 thermocycler (Eppendorf, Hamburg, Germany) and the reaction mixture (25 μL) contained template DNA (5 ng), dNTP mix (400 nM of each the deoxynucleotides), MgCl_2_ (2 mM), Kapa High Fidelity polymerase (0.5 U), 5·x·High Fidelity buffer (1×), and specific primers (400 nM of each of the two primers) targeting the V3 - V4 bacterial variable region of 16S rDNA (16S-V3-F: 5′TCGTCGGCAGCGTCAGATGTGTATAAGAGACAGCCTACGGCWGCAG 3′ and 16S-V4-R: 5′GTCTCGTGGGCTCGGAGATGTGTATAAGAGACAGGACTACHVGGGTATCTAATCC3′) or fungal ITS2 variable region (ITS2-F: 5′TCGTCGGCAGCGTCAGATGTGTATAAGAGACAGGATGAAGAAGGAGAGARAA3’ and ITS2-R: 5′GTCTCGTGGGCTCGGAGATGTGTATAAGAGACAGTCCTCCGCTTATTGATATGC3’). The primers used enable obtaining the optimal size of a PCR product for sequencing with Illumina MiSeq technology. Furthermore, at the 5′ end of the primers, Illumina Nextera XT adapter (Illumina, San Diego, CA, USA) overhang nucleotide sequences were included in the sequences of gene-specific primers, allowing subsequent sample preparation with Nextera XT DNA Sample Preparation Kit (Illumina). 

Prior to the preparation of the sequencing libraries, the obtained PCR products were analyzed by horizontal gel electrophoresis (1.5% agarose with ethidium bromide in 1×·TAE). Three technical repeats of PCR reactions per sample were combined and then purified using Agencourt^®^ AMpure XP bead (Beckmann Coulter, Brea, CA, USA). Sequencing libraries were then prepared using the Nextera XT DNA Sample Preparation Kit (Illumina), according to the manufacturer’s protocols. Samples were re-purified using Agencourt^®^ AMpure XP bead (Beckmann Coulter). The correctness of libraries, their quality and concentration were verified with electrophoresis in a 1.5% agarose gel, Qubit dsDNA HS Assay and Qubit Fluorometer (Invitrogen, Merelbeke, Belgium), and with KAPA Library Quantification Kits (Illumina). The sequencing of the amplicons was carried out using MiSeq genomic sequencer and MiSeq Reagent kit v3 kit in, 600 cycles (Illumina) with a read length of 2·×·300 nucleotides.

### 2.6. Bioinformatic Processing of the Sequenced Data

Computational analyses of the data obtained were performed using a local computing environment with the Quantitative Insights in Microbial Ecology (QIIME2, v2018.11) [10]. Briefly, raw sequences were processed with Cutadapt software enabling the trimming of the nucleotides corresponding to the sequences of primers and adapters used for PCR amplification and library preparation. In the next step, sequences were imported into the QIIME2 environment and then denoised with the use of the dada2 pipeline [11]. After obtaining feature tables and representative sequences, taxonomy classification was performed based on the SILVA v132 database (https://www.arb-silva.de/) [12] for bacteria and UNITE-v8 (https://unite.ut.ee/) [13] for fungi analyses. In the case of bacterial amplicons, additional filtering was performed in order to exclude unassigned and unspecific sequences corresponding to mitochondria, chloroplasts, and archaea as well as the sequences assigned only to a kingdom-level category of Bacteria, without any lower assignments (sequences probably associated with mitochondrial DNA). Raw sequences obtained in this study were deposited in the Sequence Read Archive (SRA; NCBI) database under the accession number PRJNA526129.

### 2.7. X-ray Powder Diffraction (XRD) Analysis

The mineral composition of the sample was examined by the XRD powder diffraction method with a Rigaku SmartLab XRD diffractometer (Rigaku Corporation, Tokyo, Japan) (copper anode, step of measurement 0.05, counting time 1 s). The main phases were identified by means of the measurement of interplanar distances and the subsequent comparison of the obtained set of data with the pattern set. Diffraction data processing was performed based on the data present in the International Center for Diffraction Data (ICDD) 2014 directory and a XRAYAN - X-ray phase analysis computer program.

### 2.8. Measurements of the Chemical Composition of Air

In the context of the chemical composition of the air inside and outside measurements of air in each season were performed using a GASMET DX-4030 (Gasmet Technologies Oy, Vantaa, Finland). In each series of measurements, readings for the concentration were performed for 25 selected gases by determining the infrared spectrum using Fourier Transformation (FT-IR). Each reading was taken for 20 s and was repeated five times. Results are shown as the average values and the standard deviation (SD) was 2%. 

### 2.9. Quantitative Analysis of Particulate Matter Concentrations in the Air

Measurements of temperature, air humidity and particulate matter (PM) concentration were performed using the TROTEC^®^ PC220 device (Trotec GmbH & Co. KG, Heinsberg, Germany). The device was calibrated in accordance with DIN EN ISO 9000 and has received the calibration certificate Deutsche Akkreditierungsstelle (DAkkS GmbH) issued by the only accreditation body in Germany operating in accordance with the German Federal Accreditation Law. The device was located 1.5 m above the ground surface, inside the Rotunda. Before each measurement, the device was left to adapt to the conditions prevailing in the building for a minimum of 5 min. The air temperature was measured in degrees Celsius (°C), and the relative humidity of the air was expressed as a percentage of water vapor in the air (%). Along with the measurement of temperature and air humidity, the device calculates the dew point temperature (DP) and the temperature of the wet thermometer (WB). The device performs a measurement of the following fractions of dust: PM2.5 and PM10. The PM abbreviation means particulate matter with an aerodynamic diameter grain size expressed in µm. A dust measurement was 1 min long and was performed in triplicate. Results are shown as average values.

### 2.10. Scanning Electron Microscopy (SEM)

Sand-like material and dust collected from ancient walls and the floor were fixed in formaldehyde vapors in a desiccator for three weeks in the presence of a silica gel desiccant. Prior to observation, the samples were coated with gold. Preparations were viewed in the LEO 1430VP scanning electron microscope (LEO Electron Microscopy, Carl Zeiss, Oberkochen, Germany) with an accelerating voltage of 20 kV.

## 3. Results

### 3.1. Environmental Conditions in the Rotunda of Sts. Felix and Adauctus

The construction of the Rotunda internal walls is presented in Figure 1a,b. The irregular structure of sandstone plates and lime mortar is clearly visible in Figure 1b.

An XRD analysis of an irregular sandstone plate (Figure 2a) showed that the main wall compound was quartz (SiO_2_) with an admixture of feldspars (potassium), plagioclase, illite, and kaolinite, while lime mortar (Figure 2b) was composed of calcite (CaCO_3_) with an admixture of feldspars (potassium), plagioclase, and quartz in a low concentration that was characteristic of mortar in the 10th/11th century in Poland.

Samples of air for chemical compound and PM concentration analysis were collected in four time points of a summer touristic season (from April to October), as the majority of tourist traffic at the castle and at the bottom of the castle hill is recorded in this period. Environmental physico-chemical parameters specific for Rotunda are summarized in Table 1 and Table 2. 

Overall, the worst conditions (expressed as the chemical parameters exceeding the reference values Directive 2008/50/EC [14] and related Polish regulations) were observed in August while the best ones were identified in June. Elevated concentrations of the following compounds were recorded most often: methane, cyclohexane, benzene, acetone, ethanol, and acetaldehyde (Table 1). Their increased concentrations persisted throughout the year Particulate matter (PM) concentrations in the air exceeded the reference values in October (86 µg/m^3^) and the increase persisted until late autumn and winter (from November to March). During the heating period in autumn and winter, PM concentrations inside the Rotunda were twice lower than those recorded in Krakow. The limited penetration of outdoor air pollutants resulted from the fact that the archaeological reserve is located in a larger building. The recorded temperature was relatively stable (a minimum of 19.3 °C in April and a maximum of 24.6 °C in August) in contrast to the humidity, which ranged from 34.3 % in April to 63 % in August. 

### 3.2. Scanning Electron Microscopy (SEM)

The scanning electron microscopy images revealed that the lime mortar was more often inhabited by microorganisms, especially fungi, than the sandstone plates (Figure 3 and Figure 4). On the surface of the sandstone plates, the traces of weathering are clearly visible (Figure 3b). The pits etched in the surface of sandstone were colonized by single bacterial cells (Figure 3c). The sandstone dust particles collected from the surface of irregular sandstone plates were covered mainly by fungi and no bacterial cells were identified in microscopic images. Lime mortar particles were inhabited by different fungi, and a *Streptomyces* pseudomycelium was observed in a microniche (Figure 4a). Numerous hyphae producing different types of spores covered most of the lime mortar particles.

### 3.3. Siderophores

Siderophore production was examined for a mixture of bacterial cells cultivated on different media and for a bacterial mixture isolated directly from a sandstone piece. The number of microorganisms after incubation in the GASN medium was 1 × 10^7^ cells/mL in both types of samples. The production of siderophores was at a similar level. Siderophore concentrations were 0. 54 mM ± 0.05 and 0.45 mM ± 0.02 for the former and the latter sample type, respectively.

### 3.4. Diversity Analysis with the Use of 16S rDNA and ITS2 Marker Gene Sequencing

We applied a metagenomic approach to characterize microorganisms residing in the Rotunda of Sts. Felix and Adauctus. The main aim of our study was to characterize the diversity of the microbiome of the ancient wall surfaces as well as airborne microbiome. Analyses were performed in QIIME 2 environment. 

Richness and diversity estimation, based on the Shannon and Pielou indices (Appendix A), yielded higher values for bacteria than for fungi. Furthermore, diversity values for ancient walls were higher than those for airborne microorganisms. Interestingly, the most even distribution of bacteria was observed in sand-like/dust samples collected from walls. On the other hand, those samples were characterized by the least evenness for fungi (Appendix A). In the case of bacteria, the most abundant and even taxa were detected in August for both ancient wall-associated and airborne microorganisms. Interestingly, in the case of fungi, the highest diversity and evenness indices values for the sand-like/dust sample and the lowest diversity and evenness indices values for the air sample were observed in the October sampling period (Appendix A). 

Overall, the results demonstrated that for the air sampled onto microbiological media only 3–6 bacterial and 3 fungal phyla were identified while in the case of samples collected from ancient walls 13–19 bacterial and 5–6 fungal phyla were detected. In both sample types there was a clear domination of three bacterial phyla, i.e., Actinobacteria, Firmicutes, and Proteobacteria, and two fungal phyla, i.e., Ascomycota and Basidiomycota (a cumulative median of 97% of all sequences). In the case of samples from ancient walls, the median number of family-level and genera-level categories was 216 and 338, respectively while 232 family-level categories and 455 genera-level categories were identified for fungi (Appendix A). As a comparison, the cultivation of airborne microorganisms only allowed the detection of 23 families and 30 genera of bacteria, and 24 families and 34 genera of fungi (Appendix A). 

Almost all microorganisms detected with the use of the cultivation-dependent method were also identified with the cultivation-independent technique. However, the relative abundance of identified microorganisms varied greatly between sample types. Moreover, only one to four taxonomic groups were dominant (>2%) in both types of samples. An overview of the identified microorganisms at lower taxonomic levels is presented in Figure 5, Figure 6, and Table 3. The analysis of Bray - Curtis dissimilarity (Appendix A) showed that the samples from wall surfaces clustered together, while air samples showed a high level of dissimilarity.

#### 3.4.1. Microbiological Diversity of Ancient Wall Surfaces

In order to characterize the natural state of the microbiome of the ancient wall, samples (mainly in the form of sand-like particles and dust) were subjected to culture-independent analyses based on directly isolated total DNA and the further amplification of marker fragments and high-throughput amplicon sequencing. An analysis of bacterial diversity (Figure 5a) showed that at the family level samples from all sampling periods were highly enriched in Pseudonocardiaceae (6.5–35.1%), Bacillaceae (12.0–30.8%), Planococcaceae (3.9–15.0%), and Micrococcaceae (2.3–4.0%), and included a substantial amount of other low abundant bacteria (16.3–26.9%). In certain periods, we observed the increased abundance of bacterial families. The sample collected in April was enriched in Geminicoccaceae (10.3%), Pseudomonadaceae (5.6%), AKYG1722 (3.4%), Sporolactobacillaceae (3.0%), Nocardioidaceae (3.0%), Trueperaceae (2.7%), and Longimicrobiaceae (2.7%). The samples from June and August were both enriched in Paenibacillaceae (4.1%; 3.9%, respectively) and Streptomycetaceae (3.2%; 3.0%). High abundances of Pseudomonadaceae (3.3%) and Rhizobiaceae (5.5%) were observed in June and August, respectively. As a comparison, higher abundances of Moraxellaceae (5.0%), Propionibacteriaceae (2.8%), Euzebyaceae (2.9%), and sequences classified to the Frankiales order (3.3%) were detected in the sample collected in October.

Interestingly, at the genus level, a high number of sequences were assigned to uncultured, unclassified or genera-unspecified categories. Most sequences within these categories were detected in the sample collected in April (41.8%), followed by samples from October (27.3%), June (23.7%), and August (15.6%). The top five most abundant genera, as calculated by ranking the median abundance across samples, were *Bacillus*, *Amycolatopsis*, *Crossiella*, *Paenisporosarcina*, and *Prauserella* (Table 3).

The taxonomic identification of *Fungi* revealed that all studied samples were dominated by Aspergillaceae (58.8–65.0%) (Figure 5b). A relatively high number of sequences in all samples was also assigned to the Capnodiales order (2.5–9.1%). A low abundance was observed for most of the other fungal family-level assignments. However some enrichment could be detected e.g., in the cases of Debaryomycetaceae (2.5%) in June; Didymellaceae (4.1%) and Pleosporaceae (2.8%) in August; Leucosporidiaceae (3.4%), Hypocreales_fam_Incertae_sedis (2.1%), Pleosporaceae (2.0%) and Mrakiaceae (2.0%) in October. Similar to bacterial diversity, most of the analyzed sequences could not be assigned to a particular genus and fell into the unidentified or genera unspecified category (April - 67.3%; June - 44.3%; August - 62.9%; October - 56.5%). A high abundance in all samples was observed only for *Aspergillus* (8.8–20.7%). Among genera with a relative abundance greater than 2% in at least one sample, *Acremonium*, *Alternaria*, *Debaryomyces*, *Leucosporidium*, *Penicillium*, and *Phialosimplex* were also identified (Table 3).

#### 3.4.2. Microbiological Diversity of Indoor Air

Diversity analyses of airborne microorganisms were performed in order to identify the “mobile” microorganisms, which could be transferred by the air. Due to the low biomass of microorganisms in the air and the consequent difficulty in obtaining a sufficient amount of total DNA and appropriate PCR products of marker genes, we decided to employ culture-dependent methods, which have often been used in studies on cultural heritage microbiomes.

A diversity analysis of airborne bacteria cultivated on agar plates revealed a high relative abundance of four families in all four sampling periods. These were Micrococcaceae (13.8–38.3%); Bacillaceae (8.4–37.3%); Moraxellaceae (9.8–21.5%), and Staphylococcaceae (3.3–19.4%) (Figure 6a). Other families enriched in a given period were Rhodobacteraceae (2.9%) in April; Enterobacteriaceae (23.0%) in June; Family XII (5.7%) and Burkholderiaceae (5.3%) in August; Pseudomonadaceae (16.8%), Planococcaceae (16.4%), Corynebacteriaceae (13.8%), Enterococcaceae (4.8%), Beijerinckiaceae (3.1%), and Burkholderiaceae (2.0%) in October. At the genus-level categories, most of the analyzed sequences were classified to particular genera. Three identified microorganisms, namely *Micrococcus*, *Bacillus*, and *Staphylococcus*, were abundant across all four periods studied (Table 3). 

The taxonomic assignment of airborne Fungi, which are easily cultivated on agar plates, revealed a high abundance of three families represented in all four time points (Figure 6b). These were Aspergillaceae (26.8–48.7%), Cladosporiaceae (6.4–18.8%), and Trichocomaceae (2.1–13.5%). Other fungal families enriched in certain periods studied were Cordycipitaceae (25.5%), Lichtheimiaceae (7.1%), and Apiosporaceae (2.3%) in April; Meruliaceae (18.2%), family unspecified Capnodiales (11.8%), Sclerotiniaceae (4.5%), Didymosphaeriaceae (3.1%), and Bolbitiaceae (2.8%) in June; Polyporaceae (10.2%), Coriolaceae (5.9%), and Pleosporaceae (2.0%) in August; and Hypocreaceae (43.7%) in October. Additionally, the Meruliaceae family was enriched in both June (18.2%) and August (13.3%) while Psathyrellaceae was abundant in both April (4.3%) and August (15,6%). At the genus level, most of the analyzed sequences were assigned to specific fungal genera and only a small number of sequences fell into the unidentified or genera-unclassified category. *Penicillium*, *Cladosporium*, and *Talaromyces* dominated across all samples (Table 3).

## 4. Discussion

In this study, we sought to characterize the microbiome associated with the remains of an excavated ancient structure built in the 10th/11th century. The abundance of microorganisms in the environment is accompanied by their substantial contribution to the biodegradation of diverse substances [15]. The unwanted degradation is called biodeterioration [1] and may affect valuable cultural heritage objects [16]. 

In our study, we analyzed the microbiome of the Rotunda of Sts. Felix and Adauctus. Its interior exhibits the characteristics of an extreme environment, primarily because of nutrient limitation, poor air exchange, unstable humidity and high salinity reflected by the presence of efflorescence on internal walls (Figure 1b and Figure 3b). Furthermore, the building’s interior is not available to tourists. Thus, the residing microorganisms are most probably specific and - to a large extent - innate, as they originate from ancient walls. Nevertheless, some introduction of microorganisms could have been possible due to excavation and conservation activities. It was shown that humans and outdoor air are the major external sources of microorganisms in indoor environments [17]. Furthermore, it seems that air - polluted with aliphatic and aromatic hydrocarbons, organic acids, aldehydes, ketones and other simple compounds, such as ammonia and carbonyl sulfide (Table 1) – is the main source of carbon, nitrogen, and sulfur for heterotrophs. These compounds can be used by bacteria for the weathering of minerals, which proceeds via proton- and/or ligand-promoted dissolution pathways under aerobic conditions. Dissolution mechanisms are mediated by the geochemically reactive organic acids and siderophores produced by bacteria [18,19]. The secretion of organic acids and siderophore/strong chelating agents is usually the response to the absence of essential nutrients [18]. As shown previously, bacterial strains isolated from a heavily-polluted underground gold mine were able to use even insoluble lead apatite (chloropyromorphite) as a source of phosphorus in phosphate-deficient environments by producing pyromorphite-solubilizing secondary metabolites [20].

In this study, we demonstrated the ability of the Rotunda’s bacterial community to produce siderophores, which indicates that the studied bacteria actively modify their surroundings. Zanardini and colleagues [21] showed that the bacterial colonization of sandstone samples from Portchester Castle (UK) was concentrated within specific sheet structures of aluminum-containing phyllosilicate minerals, most likely glauconite. Whereas sandstone plates in the Rotunda of Sts. Felix and Adauctus also contain phyllosilicate minerals, these are kaolinite and illite. As in the study of Zanardini et al. [21] we found bacterial cells located in pores and cavities (Figure 3c) and similarly to their findings, species belonging to *Bacillus* and *Arthrobacter* were one of the most abundant genera identified on the sandstone walls of the Rotunda.

The analyses of the Rotunda microbiome were carried out with the use of the high-throughput sequencing of commonly acknowledged diversity markers –16S rDNA gene and ITS fragments that have been successfully employed previously in the characterization of museum and historic buildings [6,22,23,24,25] and for many years have routinely been used in microbiome studies of biogas systems [26,27,28], aquatic habitats [29,30], soils [31,32], intestinal tracts [33,34], and many other environments [35,36].

In our study, we analyzed the microbiome associated with two habitats: (1) the wall surfaces and (2) the air within the Rotunda building. Culture-independent methods were preferentially applied, as suggested by Laiz et al. [37]. However, due to the low biomass of microorganisms in the air, culture-dependent methods were also used. Direct microbiome analysis enables the determination of the native structure of a microbial community, while cultivation experiments allow addressing the questions concerning easily cultivated, spore-forming microorganisms [37]. Previously, Dyda et al. [25] showed that a combination of different media increased the probability of obtaining a broad range of microorganisms, especially with the use of Wort agar.

The diversity analysis of bacteria and fungi associated with the ancient Rotunda of Sts. Felix and Adauctus showed that a substantial number of marker gene fragments did not share homologous sequences with the gene sequences stored in databases and could not be assigned to the genus level. This indicates that the microbiome of the Rotunda walls is largely endemic and that yet unknown microorganisms form the community structure. Nevertheless, the results demonstrate that biodetorioration could be mediated by Actinobacteria, Firmicutes, and Proteobacteria as well as Ascomycota and Basidiomycota. Numerous microorganisms from these phyla (such as Pseudomonadaceae and Aspergillaceae detected in our samples) were shown to produce deteriorating agents, such as organic and inorganic acids, chelating agents, extracellular enzymes, and polymeric substances [2,3,38,39,40]. These compounds can solubilize or change the structure of the mineral components of stone. This observation is especially important for this study, as sandstones and their main components - feldspares and silica –could be relatively easily transformed from a crystalline to an amorphous form. One evidence of the process in the Rotunda of Sts. Felix and Adauctus is the presence of sand-like material and dust collected from the surface of an irregular sandstone plate. Moreover, in situ microscopic observations confirmed the presence of fungal and bacterial cells on the deteriorated fragments of masonry and mortar, which suggests the direct contribution of microorganisms to the biodetorioration process.

It seems that fungi could have a more negative impact because of both their high ability to penetrate into masonry and mortar and the acidification of building materials previously demonstrated for the representatives of some genera, such as *Aspergillus* or *Penicillium* [2,3,41]. Unstable humidity promotes mechanical damage caused by the alternate contraction and expansion of fungal mycelia. Fungi are also known to destroy monumental stones by the secretion of organic acids, such as oxalic acid and citric acid, which exhibit chelating properties and increase the solubility of mineral cations (e.g., Ca, K and Al) present in the matrix surface [18,19]. Various fungal genera found in our study (including *Cladosporium*, *Aspergillus*, and *Alternaria*) were previously reported as deteriorating agents (e.g., in the gate of the Cathedral of Huesca in Spain) [19].

Alongside the biodeterioration risk, fungi pose a serious threat to human health [42,43], especially for immunocompromised people. Many of the fungi identified in our study (e.g., *Aspergillus*, *Penicillium*, *Alternaria*, *Cladosporium*, and *Talaromyces*) are considered strong allergens and potential human pathogens. Here, they were mostly detected in the air sampled onto agar plates. Similarly, many of the airborne bacteria identified in cultivation-based experiments (*Pseudomonas*, *Streptococcus*, *Staphylococcus*, *Micrococcus*, *Kocuria*) may potentially pose health threats [44,45]. The predominance of the potentially pathogenic microorganisms in the air may confirm that humans are the main source of indoor pollution [17].

Most of the microorganisms detected on agar plates were not predominant in the sand-like/dust samples collected from the surfaces of ancient walls. Few microbes were predominant (>2%) in both sample types. It should be noted, however, that cultivation-based methods are not optimal for diversity analyses, as they induce the selection of microorganisms. Moreover, the vast majority of environmental microbial isolates will not grow under laboratory conditions, while those that may be cultivated often enter a viable but non-culturable state in response to stress [46,47]. Interestingly, in both sand-like/dust and air samples, one of the most common and predominant microorganisms were Bacillaceae strains. Their abundance on the surfaces of ancient walls may be especially significant for the preservation of a building structure as they may be considered responsible for the self-healing of masonry and mortar. For example, species of *Bacillus* and *Paenisporosarcina* (relatively abundant in the tested samples) are known for their ability to induce the precipitation of minerals, especially calcite [48]. This in turn may lead to the closing of cracks and thus the reduction of stone penetration and contamination by destructive microorganisms. Furthermore, bacterial diversity analysis revealed that *Amycolatopsis* was abundant on the surfaces of the Rotunda walls. The species of the *Amycolatopsis* genus are able to produce various antibiotics and other metabolites with antimicrobial properties [49,50]. Its presence could result in the reduction of microbial biomass, including the microbes responsible for biodeterioration.

The enrichment of species belonging to *Bacillus* or *Amycolatopsis* could be somewhat related to the halophilic conditions at the wall surfaces caused by the salt efflorescence [50,51]. Furthermore, it must be noted that during the analysis of 16S rDNA fragments (supposedly specific for bacteria), we also detected numerous sequences of halophilic archeons (Appendix A). Their number accounted for up to 23% of all sequenced data. However, for the consistency of the bacterial amplicon analysis they were filtered out. Nevertheless, this result showed that halophilic microorganisms are abundant on the surfaces of masonry and mortar and should also be carefully studied as they can contribute to the deterioration of historic buildings [52].

Optimally, the estimation of biodeterioration risk should include an analysis of both biotic and abiotic factors. However, the significance of physico-chemical and biological factors and their interconnectedness are difficult to estimate. In our study, there was no clear indication as to the the factors affecting microbial structure shifts in the studied time periods. Previous reports demonstrated that hydrocarbons adsorbed on historical buildings could promote the growth of microorganisms and thus increase the deterioration risk [53]. Additionally, the chemoautotrophic bacteria that obtain energy by oxidizing sulfur and nitrogen produce sulfuric and nitric acids that may cause the deterioration of acid-sensitive materials, such as limestone and concrete [3,4].

Nowadays, thanks to the development of molecular methods, the most promising approach to the assessment of microbial interactions and the impact of environmental factors is to use “omic” technologies [54].

## Figures and Tables

**Figure 1 microorganisms-07-00416-f001:**
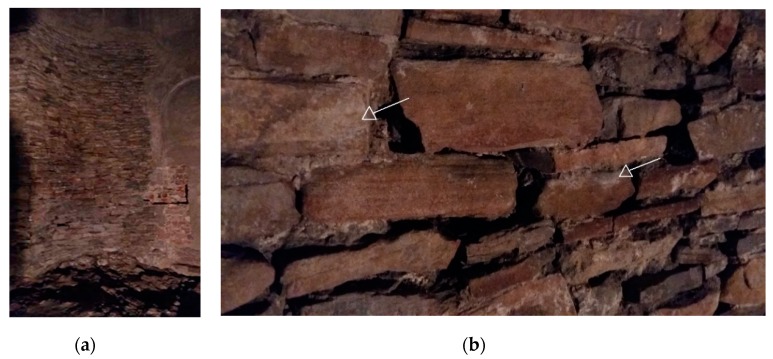
An internal wall of Sts. Felix and Adauctus Rotunda: (**a**) General view of the highest preserved wall part; (**b**) Zoomed view of the structure of irregular sandstone plate bound with lime mortar. Numerous cavities and crevices as well as efflorescence are clearly visible (marked by arrows).

**Figure 2 microorganisms-07-00416-f002:**
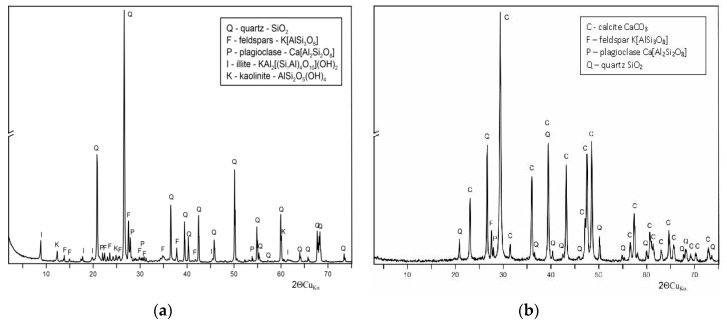
X-ray powder diffraction (XRD) analysis of (**a**) sandstone plate; (**b**) lime mortar.

**Figure 3 microorganisms-07-00416-f003:**
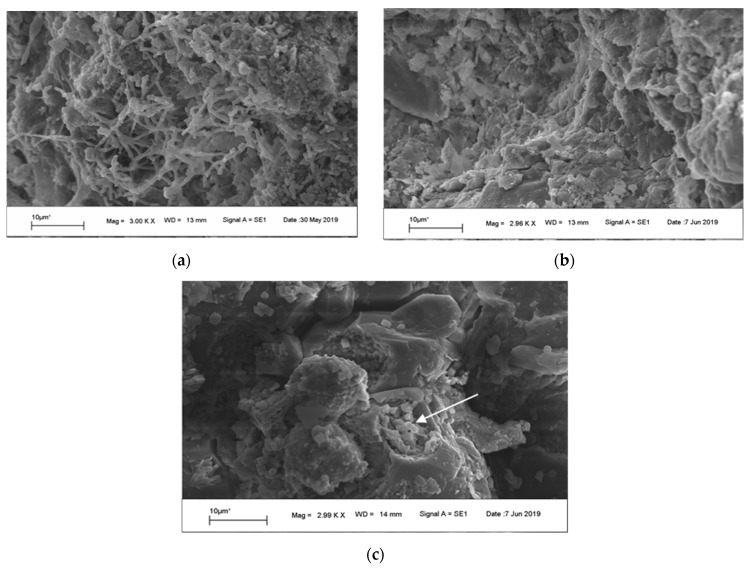
Scanning Electron Microscopy (SEM) images of sandstone samples: (**a**) the surface of sandstone dust covered with hyphae; (**b**) the weathered surface of a sandstone plate; (**c**) single bacterial cells located in the pit etched in the surface of sandstone (arrow).

**Figure 4 microorganisms-07-00416-f004:**
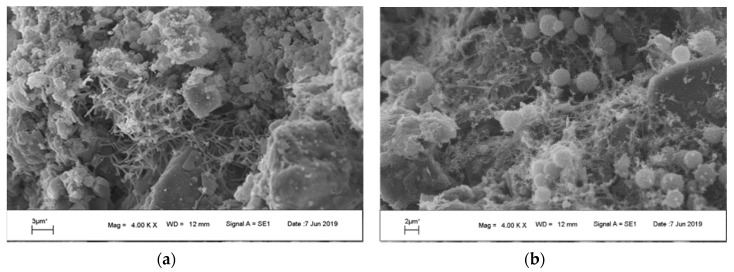
SEM images of lime mortar pieces: (**a**) pseudomycelium of *Streptomyces*; (**b**) hyphae with spores.

**Figure 5 microorganisms-07-00416-f005:**
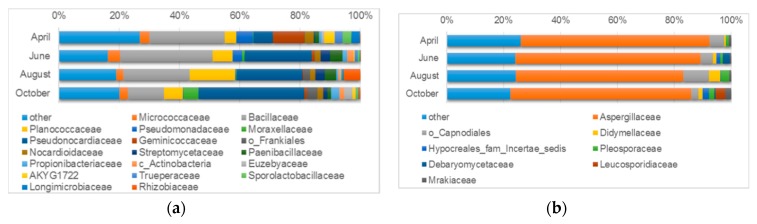
Microbial diversity of ancient walls based on deeply sequenced amplicons covering: (**a**) the bacterial V3V4 16S rDNA gene fragment and (**b**) the fungal ITS2 (internal transcribed spacer 2) region. The bar chart shows by default the relative abundance of families with an abundance greater than 2% in at least one variant. Sequences that were not assigned at the family level were named in accordance with the lowest available taxonomy: o - order, c - class.

**Figure 6 microorganisms-07-00416-f006:**
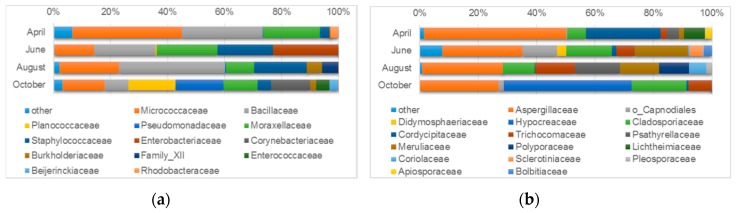
Microbial diversity of indoor air based on deeply sequenced amplicons covering: (**a**) the bacterial V3V4 16S rDNA gene fragment and (**b**) the fungal ITS2 region. The bar chart shows by default the relative abundance of bacterial families with an abundance greater than 2% in at least one variant. Sequences that were not assigned at the family level were named in accordance with the lowest available taxonomy: o - order.

**Table 1 microorganisms-07-00416-t001:** Concentrations of selected compounds (mg/m^3^) in the air of the Rotunda of Sts. Felix and Adauctus.

Compound	April	June	August	October
carbon dioxide	0.14	0,12	0.12	0.16
nitric oxide (NO)	0.00	0.25	5.27	0.00
acetic acid	0.00	0.00	0.06	1.75
methane	1.59	1.05	0.73	1.90
sulfur dioxide	0.00	0.00	0.00	0.95
ammonia	0.00	0.02	0.02	0.04
carbonyl sulfide	0.00	1.47	1.51	0.00
hydrogen cyanide	0.00	1.10	0.97	0.00
cyclohexane	0.07	0.26	0.21	0.34
formaldehyde	0.16	0.04	0.08	0.00
benzene	0.83	0.68	5.21	1.85
toluene	0.11	0.00	0.00	0.42
m-xylene	0.00	0.00	0.00	0.00
acetone	1.04	0.51	1.25	0.70
methyl isobutyl ketone	0.00	0.00	0.00	0.00
ethane	0.56	0.00	0.41	0.00
ethanol	0.87	1.28	0.85	0.73
acetaldehyde	0.50	0.41	0.51	1.82
diethyl ether	0.06	0.07	0.14	0.00
1,2,4-trimetylbenzene	0.39	0.50	0.00	0.00
octane	0.00	0.48	0.54	0.33
ethylene	0.00	0.42	0.04	0.23
1,3-butadiene	1.46	0.00	1.03	0.50
styrene	2.55	0.00	0.00	0.00
cumene	0.00	0.28	0.15	0.24

**Table 2 microorganisms-07-00416-t002:** Physico-chemical conditions in the Rotunda of Sts. Felix and Adauctus. The PM abbreviation means particulate matter with an aerodynamic diameter grain size expressed in µm.

Month	Temperature (°C)	Dew Point (°C)	Wet Bulb (°C)	Humidity (%)	PM 2.5 (µg/m^3^)	PM 10 (µg/m^3^)
April	19.3	7.9	15.2	34.3	7 ± 0%	21 ± 3%
June	21.3	12	17.7	47.3	4 ± 12%	20 ± 4%
August	24.6	18.6	20.6	63	9 ± 7%	22 ± 2%
October	22.1	12.5	16.8	51.3	36 ± 4%	86 ± 3%

**Table 3 microorganisms-07-00416-t003:** The most abundant sequences (>2%) assigned to microorganisms at the genera level and cumulatively assigned to uncultured, unclassified, unidentified or unspecified categories.

Sample		April	June	August	October
Sand-like/dust samples from ancient walls	Bacteria	*Bacillus* (20.2%) *Pseudomonas* (5.6%) *Virgibacillus* (3.2%) *Crossiella* (3.1%) *Paenisporosarcina* (3.1%) *Truepera* (2.7%) *Arthrobacter* (2.7%) Genera uncultured or unspecified (41.8%)	*Bacillus* (19.7%) *Crossiella* (9.8%) *Amycolatopsis* (5.5%) *Paenisporosarcina* (3.4%) *Pseudomonas* (3.3%) *Streptomyces* (3.2%) *Virgibacillus* (3.1%) *Prauserella* (2.9%) *Arthrobacter* (2.9%) *Paenibacillus* (2.1%) *Planococcus* (2.1%) Genera uncultured or unspecified (23.7%)	*Bacillus* (14.9%) *Paenisporosarcina* (14.0%) *Crossiella* (9.6%) *Mesorhizobium* (5.4%) *Amycolatopsis* (4.3%) *Virgibacillus* (4.0%) *Prauserella* (3.9%) *Streptomyces* (3.0%) *Saccharopolyspora* (2.2%) Genera uncultured or unspecified (15.6%)	*Crossiella* (14.5%) *Amycolatopsis* (10.9%) *Bacillus* (10.3%) *Prauserella* (5.2%) *Acinetobacter* (5.1%) *Paenisporosarcina* (3.1%) Genera uncultured or unspecified (27.3%)
	Fungi	*Aspergillus* (8.8%) Genera unidentified or unspecified (67.3%)	*Aspergillus* (20.7%) *Penicillium* (7.0%) *Phialosimplex* (2.7%) *Debaryomyces* (2.5%) Genera unidentified or unspecified (44.3%)	*Aspergillus* (12.1%) *Alternaria* (2.6%) Genera unidentified or unspecified (62.9%)	*Aspergillus* (10.3%) *Phialosimplex* (4.3%) *Leucosporidium* (3.4%) *Acremonium* (2.1%) Genera unidentified or unspecified (56.5%)
Air sampled onto agar plates	Bacteria	*Bacillus* (28.4%) *Psychrobacter* (19.9%) *Pseudarthrobacter* (16.2%) *Micrococcus* (14.5%) *Arthrobacter* (3.7%) *Kocuria* (3.4%) *Staphylococcus* (3.3%) *Paracoccus* (2.2%) Genera uncultured or unspecified (0.8%)	*Bacillus* (21.4%) *Staphylococcus* (19.4%) *Enhydrobacter* (10.9%) *Acinetobacter* (10.6%) *Micrococcus* (7.7%) *Kocuria* (5.5%) Genera uncultured or unspecified (23.2%)	*Bacillus* (37.3%) *Micrococcus* (20.1%) *Staphylococcus* (18.5%) *Acinetobacter* (8.2%) *Exiguobacterium* (5.7%) *Massilia* (5.3%) Genera uncultured or unspecified (0.2%)	*Pseudomonas* (16.8%) *Corynebacterium* 1 (13.8%) *Micrococcus* (12.2%) *Acinetobacter* (10.7%) *Bacillus* (8.4%) *Lysinibacillus* (5.8%) *Staphylococcus* (4.8%) *Enterococcus* (4.8%) *Kocuria* (2.5%) *Microvirga* (2.2%) *Massilia* (2.0%) Genera uncultured or unspecified (10.8%)
	Fungi	*Penicillium* (41.9%) *Beauveria* (25.5%) *Cladosporium* (6.4%) *Circinella* (7.1%) *Aspergillus* (6.9%) *Coprinellus* (4.3%) *Talaromyces* (2.1%) Genera unidentified or unspecified (3.0%)	*Penicillium* (22.1%) *Cladosporium* (15.8%) *Porostereum* (11.8%) *Talaromyces* (6.1%) *Aspergillus* (5.3%) *Bjerkandera* (5.9%) *Botrytis* (4.5%) *Paraconiothyrium* (3.1%) *Agrocybe* (2.8%) Genera unidentified or unspecified (12.7%)	*Penicillium* (20.6%) *Coprinellus* (15.6%) *Talaromyces* (13.5%) *Cladosporium* (11.0%) *Porostereum* (10.5%) *Coriolopsis* (9.8%) *Aspergillus* (6.9%) *Trametes* (5.9%) *Bjerkandera* (2.7%) *Alternaria* (2.0%) Genera unidentified or unspecified (0.8%)	*Trichoderma* (43.7%) *Penicillium* (26.8%) *Cladosporium* (18.8%) *Talaromyces* (8.0%) Genera unidentified or unspecified (2.7%)

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
