# Peer review of "Bacterial and Fungal Diversity Inside the Medieval Building Constructed with Sandstone Plates and Lime Mortar as an Example of the Microbial Colonization of a Nutrient-Limited Extreme Environment (Wawel Royal Castle, Krakow, Poland)"

_microorganisms, 2019, doi:10.3390/microorganisms7100416_

Round 1

Reviewer 1 Report

Manuscript (microorganisms-606030) "Bacterial and fungal diversity inside the medieval building constructed with sandstone plates and lime mortar as an example of microbial colonization of an extreme environment (Wawel Royal Castle, Krakow, Poland)" analyzes cultivable and non-cultivable microbial diversity of an ancient architectonic structure of the Rotunda of Sts. Felix and Adauctus through the use of high-throughput sequencing of marker genes corresponding to fragments of 16S rDNA (for Bacteria) and ITS2 (for Fungi), all in the context of their role in the complex process of biodeterioration. As such, this manuscript is highly interesting, of interest to the readership, and I endorse its publication in the journal after some minor corrections are made to the manuscript.

General remarks:

I do not agree with the authors that the substrate that was the object of their investigation can be considered strictly extreme environment. They base that claim on the fact that it is nutrient limited. Stone in the natural environment is usually considered as an extreme but due to the effects of many adverse factors. However, in my opinion object in question lacks other defining characteristics of an extreme environment (such as high UV radiation, extreme temperature fluctuations...) as it is protected from outdoor environment. I suggest to the authors to consider changing the title of the manuscript to "...nutrient-limited extreme environment..." to reflect this.

Throughout the manuscript the authors are referring to fungal and bacterial genera as living entities (e.g. ...Bacillus, Paenisporosarcina and Amycolatopsis can positively affect...). As genera are not living entities and thus can not be found in the nature, only species can, the correct way to express this is: species of genera Bacillus, ..., can positively affect... ). Correct this throughout the manuscript.

Only species and genera names are written in italic, while higher taxonomic ranks, such as orders and families (most prominent in the manuscript) are not. Please correct this throughout the manuscript.

Specific remarks:

line 17: replace "heritage object" with "heritage objects"

line 17: replace "Degradation" with "Deterioration", as from the context of the sentence it can be concluded that you are talking about deterioration not degradation

line 22: replace "tourist" with "tourists"

line 24: replace "biodiversity" with "diversity". Since its microbial diversity it is implied that its (bio)diversity

keywords: remove bacterial 16S rDNA and fungal ITS2

line 40: replace "were identified" with "processes are known"

lines 42-43: Algae are not a taxonomic group. Either remove them of change "taxonomic groups" to "microbial groups" for example.

line 49: I would explain here that "complex interspecies consortium" that is mentioned is actually biofilm.

lines 58-59: remove explanation of what is ITS2. Its a common knowledge at this point.

Subsection 2.1: It would be nice to add a photo of studied site (The Rotunda of Sts. Felix and Adauctus). Also, add reference from where the description of the studied site was taken.

line 80: replace "material" with "materials"

line 82: replace "swabs" with "sterile swabs"

line 83: replace "sites" with "points"

line 96: replace "148h" with "7 days"

Figure 1.: consider marking saline efflorescence on the figure with arrow(s) to help readers better visualize it

line 237: inset "by" between "mainly" and "fungi"

line 239: replace "mycelia" with "hyphae"

lines 258-261: Remove complete sentence "Identification of...". This was already mentioned in the Materials and Methods section and it has no place in the Results section

Table 3: Categories "Fungi" and "Bacteria" in the "Sample" column should not be written in italic

lines 385-386: Since siderophore production was only tested with bacterial isolates replace "microbial" and "microorganisms" with "bacterial" and "bacteria", respectively

line 425: replace "fungal microorganisms" with "fungi"

line 434-440: Though they can indeed be potentially pathogenic to humans (if right conditions are met), species of the mentioned fungal genera are actually mainly allergenic. Please comment on this in the discussion section.

Author Response

Response to Reviewer 1 Comments

Thank you very much for all your comments and remarks.
Please find our response attached below.

General remarks:

Point 1: I do not agree with the authors that the substrate that was the object of their investigation can be considered strictly extreme environment. They base that claim on the fact that it is nutrient limited. Stone in the natural environment is usually considered as an extreme but due to the effects of many adverse factors. However, in my opinion object in question lacks other defining characteristics of an extreme environment (such as high UV radiation, extreme temperature fluctuations...) as it is protected from outdoor environment. I suggest to the authors to consider changing the title of the manuscript to "...nutrient-limited extreme environment..." to reflect this.

 Response 1: We agree with the Reviewer and the title was modified as follow: “Bacterial and fungal diversity inside the medieval building constructed with sandstone plates and lime mortar as an example of microbial colonization of nutrient-limited extreme environment (Wawel Royal Castle, Krakow, Poland)”

Point 2: Throughout the manuscript the authors are referring to fungal and bacterial genera as living entities (e.g. ...Bacillus, Paenisporosarcina and Amycolatopsis can positively affect...). As genera are not living entities and thus can not be found in the nature, only species can, the correct way to express this is: species of genera Bacillus, ..., can positively affect... ). Correct this throughout the manuscript.

 Response 2: As suggested, text has been corrected throughout the manuscript. (lines: 28; 391; 450; 454, 458) including description of Table 3.

Point 3: Only species and genera names are written in italic, while higher taxonomic ranks, such as orders and families (most prominent in the manuscript) are not. Please correct this throughout the manuscript

 Response 3: As suggested, the text has been corrected (lines: 272-273, 290-291, 294 – 300, Table 3, 315 – 316, 318 – 320, 333 – 338, 350 – 356, 413 – 415, 449).

Specific remarks:

 Point 1: line 17: replace "heritage object" with "heritage objects

Response 1: As suggested, the sentence has been corrected.

Point 2: line 17: replace "Degradation" with "Deterioration", as from the context of the sentence it can be concluded that you are talking about deterioration not degradation

Response 2: As suggested, the sentence has been corrected.

Point 3: line 22: replace "tourist" with "tourists"

Response 3: As suggested, the sentence has been corrected.

Point 4: line 24: replace "biodiversity" with "diversity". Since its microbial diversity it is implied that its (bio)diversity

Response 4: As suggested, the sentence has been corrected.

Point 5: keywords: remove bacterial 16S rDNA and fungal ITS2

Response 5: As suggested, these words were removed from the keywords list

Point 6: line 40: replace "were identified" with "processes are known"

Response 6: As suggested, the sentence has been corrected.

Point 7: lines 42-43: Algae are not a taxonomic group. Either remove them of change "taxonomic groups" to "microbial groups" for example.

Response 7: As suggested, the sentence has been corrected (Algae were removed).

Point 8: line 49: I would explain here that "complex interspecies consortium" that is mentioned is actually biofilm.

Response 8: It was changed according Reviewer remarks.

Point 9: lines 58-59: remove explanation of what is ITS2. Its a common knowledge at this point.

Response 9: It was removed.

Point 10: Subsection 2.1: It would be nice to add a photo of studied site (The Rotunda of Sts. Felix and Adauctusng). Also, add reference from where the description of the studied site was taken.

Response 10: The photo of the studied site is presented as Figure 1 in Subsection 3.1. Description of the studied site was made by authors using the information from official website of Wawel Castle (websites addresses were completed in the text – lines 66-67 and 77).

Point 11: line 80: replace "material" with "materials"

Response 11: As suggested, the sentence has been corrected.

Point 12: line 82: replace "swabs" with "sterile swabs"

Response 12: As suggested, the sentence has been corrected.

Point 13: line 83: replace "sites" with "points"

Response 13: As suggested, the word has been corrected.

Point 14: line 96: replace "148h" with "7 days"

Response 14: As suggested, it has been corrected.

Point 15: Figure 1.: consider marking saline efflorescence on the figure with arrow(s) to help readers better visualize it

Response 15: It was done.

Point 16: line 237: inset "by" between "mainly" and "fungi"

Response 16: As suggested, the sentence has been corrected.

Point 17: line 239: replace "mycelia" with "hyphae"

Response 17: As suggested, the sentence has been corrected in the text and description of figures 3 and 4.

Point 18: lines 258-261: Remove complete sentence "Identification of...". This was already mentioned in the Materials and Methods section and it has no place in the Results section

Response 18: As suggested, the sentence has been removed.

Point 19: Table 3: Categories "Fungi" and "Bacteria" in the "Sample" column should not be written in italic

Response 19: As suggested, it has been corrected.

Point 20: lines 385-386: Since siderophore production was only tested with bacterial isolates replace "microbial" and "microorganisms" with "bacterial" and "bacteria", respectively

Response 20: As suggested, the sentence has been improved.

Point 21: line 425: replace "fungal microorganisms" with "fungi"

Response 21: As suggested, it was replaced.

Point 22: line 434-440: Though they can indeed be potentially pathogenic to humans (if right conditions are met), species of the mentioned fungal genera are actually mainly allergenic. Please comment on this in the discussion section.

Response 22: To clarify this, the text was modified as follow: (“Discussion”; lines 434 – 441) “Alongside the biodeterioration risk, fungi pose a serious threat to human health [42, 43], especially for immunocompromised people. Many of the fungi identified in our study (e. g. Aspergillus, Penicillium, Alternaria, Cladosporium, or Talaromyces) are considered strong allergens and potential human pathogens. Here, they were mostly detected in the air sampled onto agar plates. Similarly, many of the airborne bacteria identified in cultivation-based experiments (Pseudomonas, Streptococcus, Staphylococcus, Micrococcus, Kocuria) may potentially pose health threats [44, 45]. The predominance of the potentially pathogenic microorganisms in the air may confirm that humans are the main source of indoor pollution [17].”

Reviewer 2 Report

Dear authors,

Your study is interesting, accurate and relevant and is obviously within the scope of the journal.

I would like to ask you why you didn’t analyse salt efflorescence on walls as you carried out XRD analysis of stone and mortar. CO2, SO2 and NOx react with H2O to give H2SO4, HNO3, HCO3 that react with CaCO3 of the mortar to get CaSO4 and Ca(NO3)2: crystallisation of gypsum and nitrate is detrimental for mortar and stone. You mentioned halophilic micro-organisms which use salts but other bacteria can also generate them.

Some minor corrections have been done in the pdf file. Moreover, in the supplementary file, Figure S1 needs to be clearer by associate colours of the two sets: keeping colours for ancient walls’ microbioms and changing those for air with the same colours than for the first set but clearer. For example: dark blue colour of April in ancient walls’ microbioms and clear blue of April in air’s microbioms.

Author Response

Response to Reviewer 2 Comments

Thank you very much for all your comments and remarks.
Please find our response attached below.

General remarks:

Point 1: I would like to ask you why you didn’t analyse salt efflorescence on walls as you carried out XRD analysis of stone and mortar. CO2, SO2 and NOx react with H2O to give H2SO4, HNO3, HCO3 that react with CaCO3 of the mortar to get CaSO4 and Ca(NO3)2: crystallisation of gypsum and nitrate is detrimental for mortar and stone.

Response 1: The conservators did not agree to take appropriate samples from the surface of sandstone plates, but please consider that CaSO4 and Ca(NO3)2 were not detected in analysed samples of mortar and sandstone. Sampling of efflorescence would require scraping from the surface of sandstone which was considered invasive. The samples of sandstone and mortar which were analysed came from “naturally” broken material and their gathering does not require any interference in the material structure.

Point 2: You mentioned halophilic micro-organisms which use salts but other bacteria can also generate them.

Response 2: We agree with the Reviewer but we discussed the subject matter widely in the “Discussion”(lines: 459-465) on the basis of the results obtained in our study and literature data.

Point 3: In the supplementary file, Figure S1 needs to be clearer by associate colours of the two sets: keeping colours for ancient walls’ microbioms and changing those for air with the same colours than for the first set but clearer. For example: dark blue colour of April in ancient walls’ microbioms and clear blue of April in air’s microbioms.

Response 3: Supplementary file, Figure S1 was corrected according Reviewer remark

Specific remarks (peer-review-5178226.v1.pdf):

Response 1: line 56 - “The culture-independent methods” was replaced by “Those methods”

Response 2: line 72 – “then” was removed from the text

Response 3: line 104 – comma was inserted

Response 4: line 198 – the word “salt” was removed according to Reviewer 1 and 2 remarks

Response 5: line 215 - the word “dust” was removed and the text was modified as follow: “Samples of air for chemical compounds and PM concentrations analysis were collected…”

Response 6: line 223 – elevated level of nitric oxide (not nitric acid) was noted in June and August only while listed compounds show elevated level in all sampling time. We have not changed this point.

Response 7: line 225 – the value for PM10 was added in brackets

Response 8 and 9: lines 402 and 406 - citations were corrected – “and colleagues” was replaced by “et al.”